# Cultivation and Nitrogen Management Practices Effect on Soil Carbon Fractions, Greenhouse Gas Emissions, and Maize Production under Dry-Land Farming System

**Honglei Ren [1,\*], Shengjun Xu [2], Fengyi Zhang [1], Mingming Sun [1] and Ruiping Zhang [1]**

[1] Heilongjiang Academy of Agriculture Sciences, Harbin 150086, China; openai@haas.cn (F.Z.);
riyue@haas.cn (M.S.); xinhui@haas.cn (R.Z.)
[2] Gansu Academy of Agricultural Sciences, Lanzhou 730070, China; xusj1001@gsagr.cn
\* Correspondence: midou@haas.cn

**Abstract:** Effective nitrogen management practices by using two cultivation techniques can improve corn productivity and soil carbon components such as soil carbon storage, microbial biomass carbon (MBC), carbon management index (CMI), and water-soluble carbon (WSC). It is essential to ensure the long-term protection of dry-land agricultural systems. However, excessive application of nitrogen fertilizer reduces the efficiency of nitrogen use and also leads to increased greenhouse gas emissions from farming soil and several other ecological problems. Therefore, we conducted field trials under two planting methods during 2019–2020: P: plastic mulching ridges; F: traditional flat planting with nitrogen management practices, i.e., 0: no nitrogen fertilizer; FN: a common nitrogen fertilizer rate for farmers of 290 kg ha$^{-1}$; ON: optimal nitrogen application rate of 230 kg ha$^{-1}$; $_{ON75\%+DCD}$: 25% reduction in optimal nitrogen fertilizer rate + dicyandiamide; $_{ON75\%+NC}$: 25% reduction in optimal nitrogen rate + nano-carbon. The results showed that compared to other treatments, the $_{PON75\%+DCD}$ treatment significantly increased soil water storage, water use efficiency (WUE), and nitrogen use efficiency (NUE) because total evapotranspiration (ET) and GHG were reduced. Under the $P_{ON75\%+DCD}$ or $P_{ON75\%+NC}$, the soil carbon storage significantly (50% or 47%) increased. The $P_{ON75\%+DCD}$ treatment is more effective in improving MBC, CMI, and WSC, although it increases gaseous carbon emissions more than all other treatments. Compared with FFN, under the $P_{ON75\%+DCD}$ treatment, the overall $CH_4$, $N_2O$, and $CO_2$ emissions are all reduced. Under the $P_{ON75\%+DCD}$ treatment, the area scale GWP (52.7%), yield scale GWP (90.3%), biomass yield (22.7%), WUE (42.6%), NUE (80.0%), and grain yield (32.1%) significantly increased compared with $F_{FN}$, which might offset the negative ecological impacts connected with climate change. The $P_{ON75\%+DCD}$ treatment can have obvious benefits in terms of increasing yield and reducing emissions. It can be recommended to ensure future food security and optimal planting and nitrogen management practices in response to climate change.

**Keywords:** nitrogen management; global warming potential; soil carbon fractions; nitrogen use efficiency; farming techniques; maize production

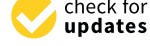



## 1. Introduction

Plastic mulching under the ridge furrow rainfall harvesting method (P) is expanding rapidly to increase rain-fed maize production in semi-arid regions [1]. From 2013 to 2019, the global demand for plastic film mulching is expected to increase by 7.6% [2]. The soil and root respiration contribute approximately 20%, 12%, and 60% of $CO_2$, $CH_4$, and $N_2O$ emissions [3]. The global carbon cycle is affected by global warming, which distorts the function and structure of ecosystems [4]. It is estimated that 65% of total $N_2O$ emissions come from soil [5], and nitrogen application accounts for 36% of direct $N_2O$ emissions from global agricultural soils [6]. In China, by 2020, reducing nitrogen input and improving water management may reduce 17% of the total greenhouse gas emissions, mainly from wheat, corn, and rice [7].

Plastic film mulching (PFM) is usually used to improve soil water storage, decrease nitrogen loss caused by leaching, provide favorable conditions for soil biological activities, and control weeds [8,9]. However, the excessive use of inorganic fertilizers in China has increased ecological problems [10], which have little influence on crop yields but have caused major nitrogen losses into the atmosphere [11]. Northwest China is an irrigated area, and numerous growers use more irrigation with unnecessary nitrogen supplies in order to raise crop production [12]. These approaches have caused severe water and nutrient deficiencies [13], decreased crop production and NUE [14], and improved the risk of GHGI [15,16]. Reducing agricultural carbon dioxide emissions can be attained by improving soil carbon sequestration [17]. Smart fertilizer management practices are essential for SOC storage [18]. Sufficient nutrients in the soil can increase biomass yield and SOC [12]. Thus, it is vital to launch more effective fertilizer management practices to use less fertilizer to increase crop yields and reduce environmental pollution.

Among various greenhouse-gas reduction strategies, fertilizers that improve NUE, such as slow-release fertilizers, can effectively reduce nitrogen loss [19]. The use of slow-release fertilizers can suspend the exchange of ammonium ($NH_4^+$) to nitrate ($NO_3^-$) by preventing nitrifying bacteria activity [20], thereby increasing the efficiency of N use, reducing $N_2O$ emissions, and maintaining or improving crop Production [21,22]. As global warming intensifies, reducing $N_2O$ emissions from agricultural soils has attracted great attention [23]. Dicyandiamide (DCD) is a highly effective nitrification inhibitor [24]. Nie et al. [25] report that the addition of DCD combined with an optimized nitrogen fertilizer rate significantly reduced $N_2O$ flux emissions by 67.3–83.8%. Nanocarbon (NC) is a new type of fertilizer synergist. Compared with urea alone, nanocarbon (NC) added to urea can increase crop production, increase nitrogen use efficiency, and reduce nitrogen loss [26]. Nanocarbon is a modified carbon with non-conductive properties and low ignition points. NC can screen poisonous gases and is currently widely used in new fertilizer research fields aimed at increasing crop yields and fertilizer utilization [10]. However, it is not clear whether nanocarbons can also provide greenhouse gas emission reduction potential, especially when compared to DCD.

Numerous researchers have focused on the effects of the separate application of NI and irrigation on greenhouse gas intensity and maize yields [27,28]. The current study aims at: (a) Estimating greenhouse gas emissions in the form of $CH_4$, $CO_2$, $N_2O$, and GWP under different fertilizer management practices; (b) Estimating SOC and microbial activities in relation to GHG emissions. (c) determine the most adaptable N management practices that provide high and stable SOC, nitrogen use efficiency, and rain-fed maize production while decreasing greenhouse gas emissions.

## 2. Materials and Methods

### 2.1. Site Location

The field trial was carried out in the 2019 and 2020 years at the Gansu Academy of Agricultural Sciences. The experimental sites are located at 103°41′17.49″ E, 36°06′3.31″ N, and 467 m asl. The rainfall from July to September exceeds 60%. The rainfall in the growing season from 2019 to 2020 was between 279 and 265 mm (Figure 1). Table 1 indicates the soil chemical properties at a depth of 20 cm. The top 0–15 cm of soil on the research site is Eum-Orthrosols (Chinese Soil Taxonomy).

**Table 1.** The chemical properties of experimental site's soil layer (0–15 cm).

| Year | pH | SOM (g kg$^{-1}$) | TP (g kg$^{-1}$) | TK (g kg$^{-1}$) | AP (mg kg$^{-1}$) | AK (mg kg$^{-1}$) |
|------|------|------|------|------|------|------|
| 2019 | 8.24 | 13.67 | 1.07 | 18.21 | 21.05 | 159.22 |
| 2020 | 8.08 | 15.33 | 1.03 | 16.34 | 18.89 | 164.65 |

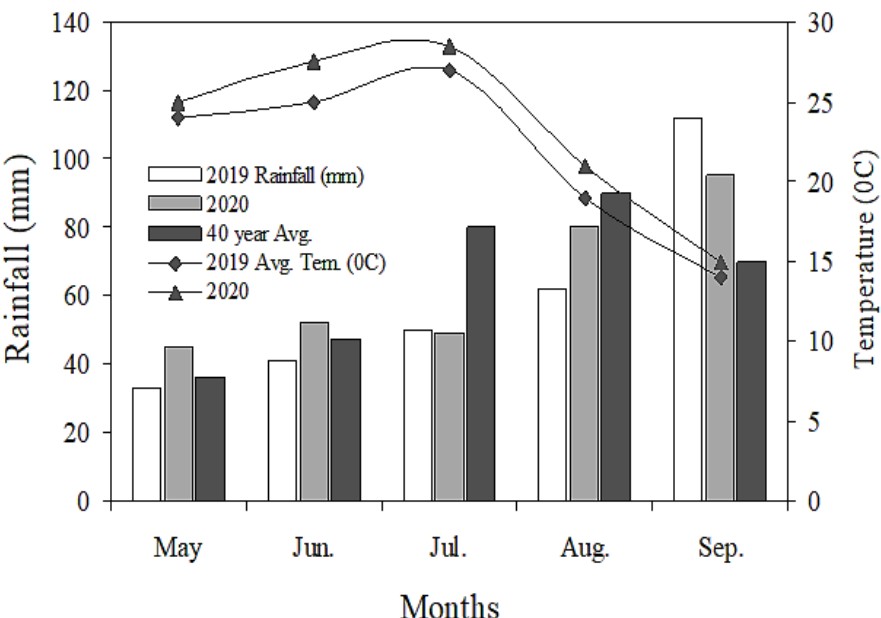

**Figure 1.** Monthly rainfall distribution during the maize-growing seasons.

*2.2. Experimental Design*

A randomized completely block design were used having three replications. The area of each plot is 60 m$^2$ (20 × 3 m$^2$). The following ten treatments and two cultivation practices P: plastic film mulching on ridges; F: traditional flat planting with five different nitrogen management practices $_0$: no N fertilizer; $_{FN}$: farmers common N rate is 290 kg ha$^{-1}$; $_{ON}$: optimal N rate is 230 kg ha$^{-1}$; $_{ON75\%+DCD}$: 25% reduction in optimal N rate + dicyandiamide (DCD) is applied at a rate of 5% of the total applied N (*w/w*); $_{ON75\%+NC}$: 25% reduction in optimal N rate + nano-carbon (NC) is applied at 0.3% (*w/w*) of the total applied fertilizer. The furrow is 60 cm wide and 15 cm high. Plant population of 75,000 ha$^{-1}$ of Dafeng 30 maize cultivar; planting time is 10 May 2019, and 9 May 2020. The corn was harvested on 10 September 2019 and 8 September 2020. In 2019–2020, weeds will be controlled by hand. The recommended doses of P and K at 90 and 60 kg ha$^{-1}$ apply one day before sowing. During both growing seasons, irrigation was not supplied, conventional tillage practices were used for soil flow, and weeds were controlled manually.

*2.3. Sampling and Measurements*

2.3.1. Soil Water Storage (mm)

Soil water storage (SWS) was determined by the following formula.

$$SWS = C \times \rho \times H \times 10 \tag{1}$$

C is the soil gravimetric moisture content (%); $\rho$ is the bulk density (g cm$^{-3}$); and H is the soil depth (0–120 cm).

2.3.2. Analysis of Gas Sampling

A cylindrical opaque chamber (inner diameter 25 cm × 20 cm height) was used. Each plot was repeated three times, and the bottom chamber was buried in the inner soil 20 cm deep. An electric fan is fixed to mix the gas. From 0 to 30 min after closing the chamber, use a 30 mL air-tight syringe to collect the gas sample with the help of a gas chromatograph equipped. A gas chromatograph equipped with an HP-PLOT Q capillary column was used to quantify the concentration of three gases ($N_2O$, $CH_4$, and $CO_2$). A flame ionization detector (FID) with a methanizer was used to analyze $CH_4$ and $CO_2$ concentrations, while the concentration of $N_2O$ was analyzed by the Ni electron capture detector.

As emission rates were determined by the equation below:

$$\text{Gas emission rate } (\text{mgm}^{-2}\text{h}^{-1}) = \Delta c/\Delta t \times V/A \times \rho \times 273/T \tag{2}$$

where $\Delta c/\Delta t$ is the difference of gas concentration between 0 and 30 min, V is the volume, A is the area, $\rho$ is the density, and T is the absolute temperature.

The seasonal gas fluxes were determined by the equation below:

$$\text{Seasonal flux } (\text{kgha}^{-1}) = \sum_{i}^{n} (R_i \times D_i) \tag{3}$$

where R is the daily gas emission rate and D is the number of days between the *i*th sampling interval.

The net GWP was determined by the equation below:

$$\begin{aligned} \text{Net GWP } (\text{kgCO}_2\text{-eq ha}^{-1}) \ = \ & \text{CH}_4 \text{ flux } 28 \\ & + \text{N}_2\text{O flux } 265 - \Delta SOC 44/12 \end{aligned} \tag{4}$$

The greenhouse gas intensity (GHGI) was determined using the net GWP per maize grain yield [3]:

$$\text{CHGI}(\text{kgCO}_2\text{-eq kg}^{-1} \text{ grain}) = \text{NetCWP/grain yield} \tag{5}$$

### 2.3.3. Global Warming Potential

The GWP for area and yield scale of income is determined by [29]:

$$\begin{aligned} \text{Area} - \text{scaled GWP} = \ & 28 \times \text{CH}_4(\text{kgha}^{-1}\text{yr}^{-1}) \\ & + 265 \times \text{N}_2\text{O}(\text{kghal}^{-1}\text{yr}^{-1}) \end{aligned} \tag{6}$$

The yield-scaled GWP was then calculated as the ratio between the area-scaled GWP and grain yield [29].

### 2.3.4. Soil Carbon Fraction Analysis

Soil MBC is determined by using a modified chloroform fumigation extraction method [30]. The mineralizable carbon (RMC) content was determined after extraction with 0.5 M $K_2SO_4$ [31], and then the soil extract was wet digested with dichromate [32]. The acid hydrolyzed carbohydrate carbon (AHC) is determined by taking the equivalent weight of 2 g of soil extracted with 20 mL of 1.5 M sulfuric acid ($H_2SO_4$) for 24 h with regular shaking and filtering through a glass fiber filter according to the procedure of [33]. The water-soluble carbohydrate carbon (WSC) content is determined by [34]. The ninhydrin reactive nitrogen (NRN) in 20-g soil samples was extracted with 0.5 M potassium sulfate ($K_2SO_4$) and estimated colorimetrically after mixing the soil extracts with ninhydrin [35].

### 2.3.5. Soil Carbon Storage, Carbon Management Index

The carbon management index (CMI) was calculated by using a reference sample value according to the procedure of Blair et al. [36]. Based on changes in between the reference and sample sites of the total carbon content, a carbon pool index (CPI) was determined by Liu et al. [37]. CPI = [sample TC/TC of reference soil].

Based on the changes in the C lability (L) = $KMnO_4$-C/TC-$KMnO_4$-C, the lability index was determined.

$$\text{LI} = [\text{sample L/reference L}]$$

$$\text{CMI} = \text{CPI} \times \text{LI} \times 100.$$

Carbon equivalent emissions (CEE) and carbon efficiency ratios (CER) were calculated using the following equations:

$$CEE = GWP \times 12/44$$

$$CER = \text{grain yield (in terms of carbon) of the maize}/CEE$$

The 43% carbon concentration in the grain was found.

### 2.3.6. Biomass and Maize Production

Biomass and grain yield of maize were measured at 6 m$^2$ area and hand harvested from each plot.

$$WUE = Y/ET \tag{7}$$

where WUE (kg ha$^{-1}$ mm$^{-1}$) is the water use efficiency, Y is the grain yield, and ET is the evapotranspiration.

Nitrogen use efficiency (NUE kg kg$^{-1}$) was calculated by Wang et al. [38].

$$NUE = GY/N \text{ uptake} \times 100\% \tag{8}$$

### 2.3.7. Statistical Analysis

Data and interactions were analyzed using an analysis of variance (ANOVA) and Analytical Software (statistic 8.1/2008/statsoft/Tulsa, OK, USA). To calculate the probability levels of P (0.05), the LSD (least significant difference) test was used.

## 3. Results

### 3.1. SWS and ET

Changes in rainfall, maize water utilization, and soil evaporation have led to reduced soil water storage (SWS) at different maize growth stages (Figure 2). In our research work, SWS showed non-significant differences among all treatments at 30 days after planting (DAP). The water consumption of maize improves the growth of plants. P$_{ON75\%+DCD}$ treatment can reduce drought and ensure the successful growth of plants. In the P$_{ON75\%+DCD}$ treatment, the SWS of maize was considerably higher than in the F$_{ON75\%+DCD}$ treatment. Start with 60–80 DAP; compared to 30 DAP, the trend of SWS for each treatment is significantly enhanced. At 100 DAP, the average data of two years shows that, compared with F$_{ON75\%+DCD}$ and F$_{ON75\%+NC}$, the SWS under the P$_{ON75\%+DCD}$ treatment is significantly the largest. The different cultivations of $_{ON75\%+DCD}$ and $_{ON75\%+NC}$ nitrogen application treatments had the largest SWS, but compared with all other treatments, the difference was considerable at various corn stages. The change in SWS was not significant between P$_{ON75\%+NC}$ and F$_{ON75\%+DCD}$ treatments at 120–140 DAP.

The corn ET is positively correlated with rainfall and nitrogen management practices. Compared with FFN and PFN treatments, P$_{ON75\%+DCD}$ and P$_{ON75\%+NC}$ treatments with different nitrogen management measures resulted in lower total ET due to high soil evaporation. The results indicated that ET at P$_{ON75\%+NC}$ treatment is considerably lower than at F$_{ON75\%+DCD}$ and F$_{ON75\%+NC}$ treatment, respectively. Regardless of the cultivation method, the $_{ON75\%+DCD}$ treatment significantly reduced 10.1% compared to the FN treatment. Compared with F$_{ON75\%+DCD}$ treatment, P$_{ON75\%+DCD}$ treatment significantly reduced ET by 7.0%, and P$_{ON75\%+NC}$ treatment significantly reduced ET by 22.8% compared with FFN treatment.

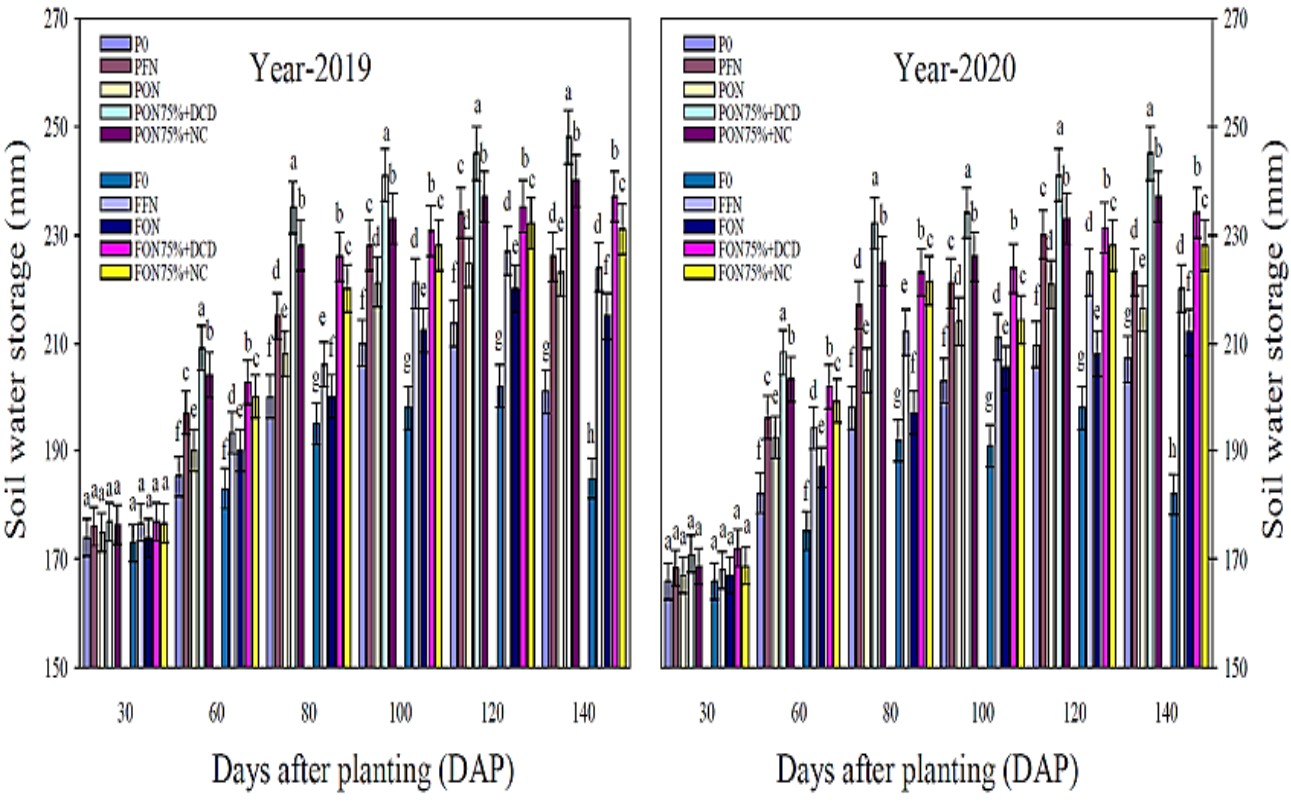

**Figure 2.** Effects of farming and nitrogen management practices on soil water storage at the depth of 0–120 cm soil layers at different growth stages of maize during 2019 and 2020. Vertical bars represent the LSD at *p* = 0.05 (*n* = 3).

*3.2. Soil Carbon Fractions*

The MBC ranges from 113.7 to 414.6 mg kg$^{-1}$ (Table 2). Under the $P_{ON75\%+DCD}$ treatment, the MBC was significantly higher (400.3 mg kg$^{-1}$) compared to the rest of all treatments. The application of $P_{ON75\%+DCD}$ treatment showed a significant increase of 67% in MBC (Table 2). Compared with other treatments, the $P_{ON75\%+DCD}$ treatment considerably improved the TC content (5.08 g kg$^{-1}$) (Table 3). The content of easily mineralizable carbon (RMC) was the highest in the plots treated with $P_{ON75\%+DCD}$ and $P_{ON75\%+NC}$ (177.7–137.9 mg kg$^{-1}$) and the lowest under the $F_0$ treatment (25.5 mg kg$^{-1}$). The WSC and AHC vary considerably under different cultivation and nitrogen management practices, ranging from 6.8 to 45.4 mg C kg$^{-1}$ and 320.9 to 583.2 mg C kg$^{-1}$. The CMI was considerably improved by 31.2%, 10.2%, 11.4%, 10.8%, 10.8%, and 13.4% under the treatments of $P_0$, $P_{FN}$, $P_{ON}$, $P_{ON75\%+DCD}$, and $P_{ON75\%+NC}$, which was higher than that of $F_0$ and $F_{FN}$, $F_{ON}$, $F_{ON75\%+DCD}$, and $F_{ON75\%+NC}$ treatments. Under different cultivation and nitrogen management measures, the total N, C:N ratio, $NH_4^+$-N, $NO_3^-$-N, and NRN have a significant impact (Tables 2 and 3). It was found that the total N under the $P_{ON75\%+DCD}$ and $P_{ON75\%+NC}$ treatments was significantly higher (0.61–0.57 g kg$^{-1}$) than that of all other treatments. Compared with the $F_{ON75\%+DCD}$ and $F_{ON75\%+NC}$ treatments, the C:N ratio was significantly higher (8.55–8.37) under the $P_{ON75\%+DCD}$ and $P_{ON75\%+NC}$ treatments. There were three peaks of $NH_4^+$-N and $NO_3^-$-N in at various growth stages during the two-year field study. The $NO_3^-$-N under $F_0$ treatment was considerably lower compared to the rest of all treatments (Figure 3). Under the treatments of $P_{ON75\%+DCD}$ and $P_{ON75\%+NC}$, ($NO_3^-$-N) significantly increased compared with $P_{FN}$ and $F_{FN}$ treatments. Compared with $F_0$, the $NH_4^+$-N of all other treatments was considerably improved, whereas the $NH_4^+$-N of the $P_{ON75\%+DCD}$ and $P_{ON75\%+NC}$ treatments did not change significantly under the two cultivation methods (Figure 4). Different cultivation and nitrogen management measures at each growth stage have a significant impact on the $NH_4^+$-N content.

**Table 2.** Soil carbon fractions and carbon management index at 0–15 cm soil depth under different cultivation and nitrogen management practices during 2019–2020 maize growing seasons.

| Treatments | MBC (mg kg$^{-1}$) | RMC (mg kg$^{-1}$) | WSC (mg kg$^{-1}$) | AHC (mg kg$^{-1}$) | CMI |
|---|---|---|---|---|---|
| **2019** | | | | | |
| $P_0$ | 171.0 g | 32.2 f | 8.1 f | 369.8 e | 88.9 e |
| $P_{FN}$ | 289.5 c | 102.3 c | 27.5 d | 474.2 c | 119.2 c |
| $P_{ON}$ | 285.1 c | 77.3 d | 26.3 d | 456.5 c | 109.5 c |
| $P_{ON75\%+DCD}$ | 386.1 a | 168.9 a | 44.7 a | 564.7 a | 142.3 a |
| $P_{ON75\%+NC}$ | 311.7 b | 130.8 b | 31.0 c | 506.0 b | 128.6 b |
| $F_0$ | 113.0 h | 22.2 g | 6.2 g | 296.4 f | 62.3 f |
| $F_{FN}$ | 213.8 e | 70.7 d | 24.5 e | 381.5 e | 105.7 d |
| $F_{ON}$ | 199.2 f | 55.9 e | 23.7 e | 370.9 e | 95.8 e |
| $F_{ON75\%+DCD}$ | 300.6 b | 116.3 b | 40.8 b | 453.8 c | 125.3 b |
| $F_{ON75\%+NC}$ | 240.2 d | 88.6 | 27.2 d | 404.9 d | 109.9 c |
| **2020** | | | | | |
| $P_0$ | 190.4 g | 35.5 f | 8.8 f | 394.2 f | 97.8 e |
| $P_{FN}$ | 314.7 d | 112.9 c | 28.6 d | 505.2 c | 123.7 c |
| $P_{ON}$ | 313.8 d | 84.5 d | 27.2 d | 485.0 d | 114.1 c |
| $P_{ON75\%+DCD}$ | 414.6 a | 186.5 a | 46.0 a | 601.7 a | 147.9 a |
| $P_{ON75\%+NC}$ | 335.5 c | 144.9 b | 32.2 c | 539.7 b | 134.8 b |
| $F_0$ | 151.7 h | 28.8 f | 7.5 f | 345.3 g | 80.1 f |
| $F_{FN}$ | 264.2 f | 91.8 d | 26.5 d | 443.3 e | 114.7 c |
| $F_{ON}$ | 256.5 f | 70.2 e | 25.4 e | 428.0 e | 105.0 d |
| $F_{ON75\%+DCD}$ | 357.6 b | 151.4 b | 43.4 b | 527.8 b | 136.6 b |
| $F_{ON75\%+NC}$ | 287.9 e | 116.7 c | 29.7 c | 472.3 d | 122.4 c |

Values are given as means, and different lowercase letters indicate significant differences at $p \leq 0.05$ levels.

**Table 3.** Soil nitrogen fractions and total carbon to total nitrogen ratio at 0–15 cm soil depth under different cultivation and nitrogen management practices during 2019–2020 maize growing seasons.

| Treatments | TC (g kg$^{-1}$) | TN (g kg$^{-1}$) | TC:TN | NRN (µg g$^{-1}$ Soil) | CEE (kg C ha$^{-1}$) | CER |
|---|---|---|---|---|---|---|
| **2019** | | | | | | |
| $P_0$ | 2.42 c | 0.47 b | 5.1 f | 3.3 e | 1846 e | 0.82 c |
| $P_{FN}$ | 3.85 b | 0.54 a | 7.1 b | 7.4 b | 2395 b | 0.94 b |
| $P_{ON}$ | 3.29 b | 0.51 a | 6.4 c | 6.9 c | 2149 d | 0.90 b |
| $P_{ON75\%+DCD}$ | 4.97 a | 0.62 a | 8.0 a | 10.5 a | 2614 a | 1.01 a |
| $P_{ON75\%+NC}$ | 4.74 a | 0.57 a | 8.3 a | 8.8 b | 2334 b | 0.95 b |
| $F_0$ | 1.74 d | 0.43 b | 4.1 g | 3.1 e | 1584 f | 0.73 d |
| $F_{FN}$ | 3.18 b | 0.46 b | 6.9 d | 4.9 d | 2171 d | 0.82 c |
| $F_{ON}$ | 2.61 c | 0.45 b | 5.9 e | 4.8 d | 2144 d | 0.81 c |
| $F_{ON75\%+DCD}$ | 4.29 a | 0.52 a | 8.3 a | 7.4 b | 2376 b | 0.92 b |
| $F_{ON75\%+NC}$ | 4.06 a | 0.49 b | 8.4 a | 6.4 c | 2258 b | 0.84 c |
| **2020** | | | | | | |
| $P_0$ | 2.64 d | 0.51 a | 5.2 d | 5.4 d | 1502 f | 0.80 b |
| $P_{FN}$ | 4.08 b | 0.54 a | 7.6 b | 7.2 c | 2190 d | 0.91 a |
| $P_{ON}$ | 3.51 c | 0.53 a | 6.7 c | 7.1 c | 2108 d | 0.88 b |
| $P_{ON75\%+DCD}$ | 5.19 a | 0.60 a | 8.7 a | 9.7 a | 2682 a | 0.97 a |
| $P_{ON75\%+NC}$ | 4.96 b | 0.57 a | 8.8 a | 8.7 b | 2471 b | 0.93 a |
| $F_0$ | 2.19 d | 0.47 b | 4.7 d | 3.9 e | 1404 g | 0.69 c |
| $F_{FN}$ | 3.63 c | 0.51 a | 7.1 b | 6.5 c | 2093 e | 0.80 b |
| $F_{ON}$ | 3.06 c | 0.49 b | 6.2 c | 6.3 c | 2010 e | 0.77 c |
| $F_{ON75\%+DCD}$ | 4.74 b | 0.58 a | 8.2 a | 9.2 a | 2284 c | 0.87 b |
| $F_{ON75\%+NC}$ | 4.51 b | 0.54 a | 8.4 a | 8.0 b | 2173 d | 0.82 b |

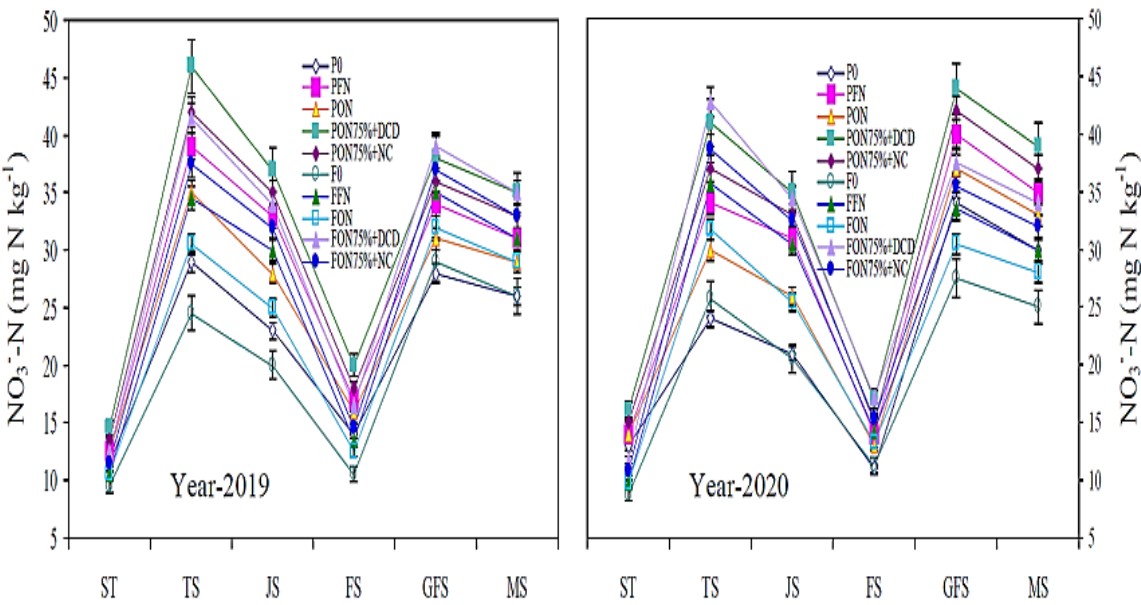

**Figure 3.** Effects of farming and nitrogen management practices on $NO_3^-$-N contents. The vertical bars represent the standard error of the mean (*n* = 3).

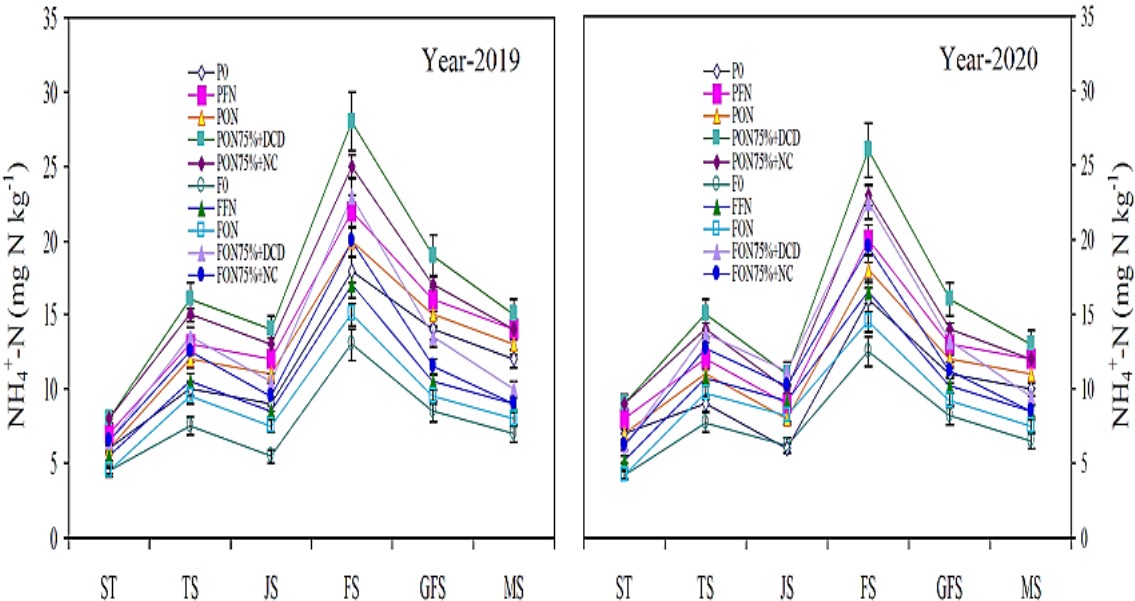

**Figure 4.** Effects of farming and nitrogen management practices on $NH_4^+$-N contents.

*3.3. Greenhouse Gas Emissions*

Field research found that $CO_2$ emissions were positive and experienced three fluctuations (Figure 5). Regardless of the different planting and nitrogen management methods, $CO_2$ is at its minimum during the sowing period, increases significantly during the flowering period, and reaches its highest during the grain filling period. Compared with FFN treatment, $P_{ON75\%+DCD}$ and $P_{ON75\%+NC}$ considerably changed $CO_2$, while the emissions of $F_{ON75\%+DCD}$ treatments were considerably higher than $F_{ON75\%+NC}$. Compared with $P_0$, $CH_4$ was considerably lower compared with the rest of the treatments, while the $CH_4$ of $_{ON75\%+DCD}$ and $_{ON75\%+NC}$ did not change considerably under the two cultivation methods (Figure 6). Different cultivation and nitrogen management practices apply to all growth stages. There were two peaks of $N_2O$ during the jointing and flowering periods. The $N_2O$ at $F_0$ is significantly lower than the rest of all treatments (Figure 7). Under $P_{ON75\%+DCD}$ and $P_{ON75\%+NC}$ treatments, $N_2O$ emissions are significantly higher than those of PFN and

FFN treatments. Under different cultivation and nitrogen management measures, $N_2O$ emissions at different growth stages have a significant impact.

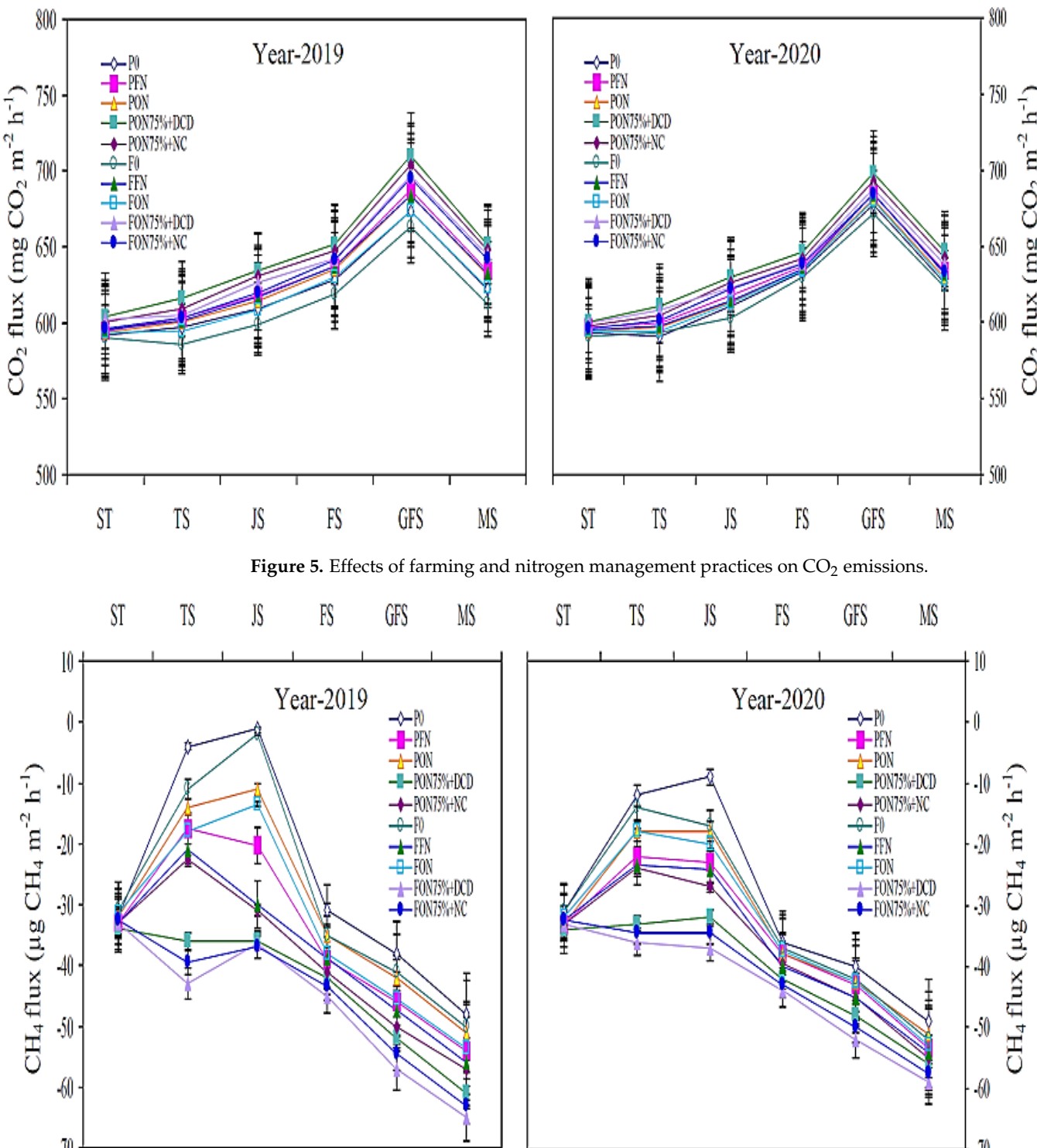

**Figure 5.** Effects of farming and nitrogen management practices on $CO_2$ emissions.

**Figure 6.** Effects of farming and nitrogen management practices on $CH_4$ emissions.

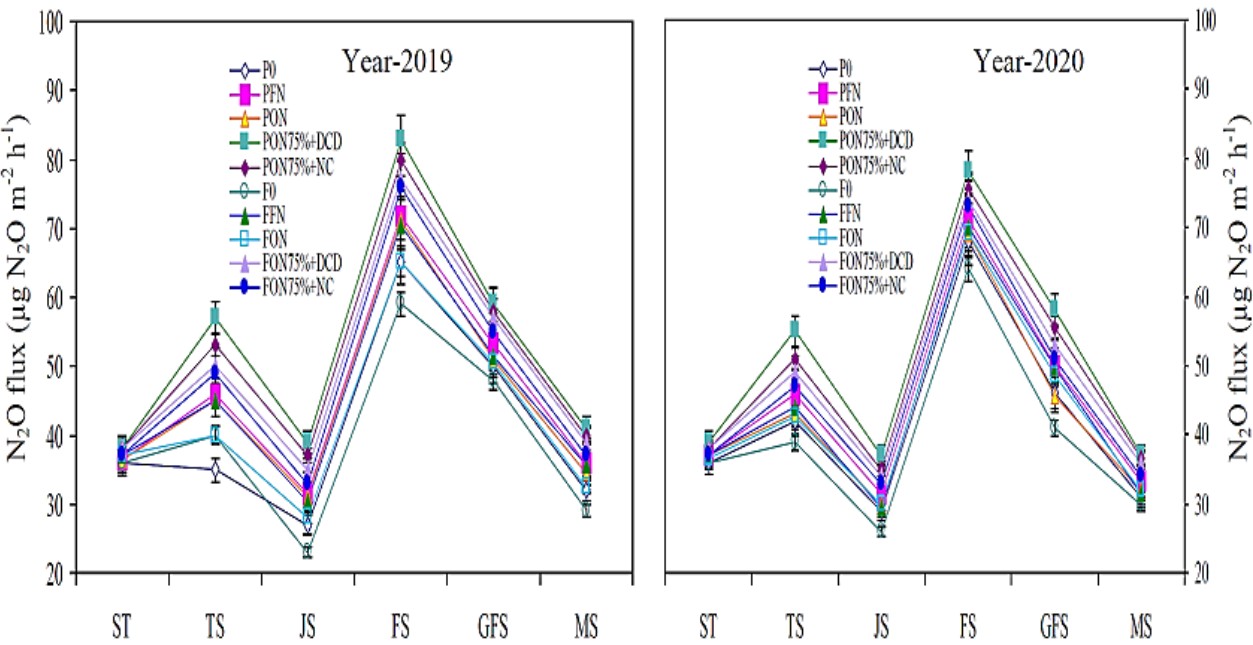

**Figure 7.** Effects of farming and nitrogen management practices on $N_2O$ emissions.

### 3.4. GWP, GHGI, and CEE

Under different cultivation and nitrogen management practices, the effects of $P_{ON75\%+DCD}$ treatments on GHGI are different, which shows the net GWP per grain yield (Table 4). Under $P_{ON75\%+DCD}$ and $P_{ON75\%+NC}$, GHGI was significantly reduced because of the substantial increase in corn production. This enhancement in GHGI is regularly influenced by soil carbon pool depletion rather than improved greenhouse gas emissions. Net GWP is determined by considering the GWP of $N_2O$ and $CH_4$ and changes in SOC. The net GWP largely depends on the depletion of the soil carbon pool and different cultivation and nitrogen management practices. Under the $P_{ON75\%+DCD}$, the net GWP is 19.1–19.0 Mg $CO_2$ eq. ha$^{-1}$, adding 17.2–17.5 and 1.7–1.6 Mg $CO_2$-eq. ha$^{-1}$) for soil carbon depletion and $N_2O$ (Table 4). Compared with the $P_{FN}$ and $F_{FN}$ treatments, the $P_{ON75\%+DCD}$ considerably improved the net GWP, which was mostly due to the significant increase in soil carbon pool consumption. The lowest CEE was measured in $F_0$ treatment (1494 kg C ha$^{-1}$). Under different cultivation and nitrogen management measures, the maximum CEE (2648 kg C ha$^{-1}$) was measured under the $P_{ON75\%+DCD}$ treatment.

### 3.5. Area and Yield-Scaled GWP

GWP shows that there are considerable differences between different cultivation and nitrogen application measures (Table 5). Under the $P_{ON75\%+DCD}$ and $P_{ON75\%+NC}$ treatments, the regional scale GWP in 2019–2020 is significantly higher than all other treatments. The mean value of the area scale GWP of $P_0$, $P_{FN}$, $P_{ON}$, $P_{ON75\%+DCD}$, $P_{ON75\%+NC}$, $F_{FN}$, $F_{ON}$, $F_{ON75\%+DCD}$, and $F_{ON75\%+NC}$ has increased by 33.5% and 55.7%, 50.5%, 65.5%, 62.6%, 27.1%, 18.9%, 54.0%, and 49.3%, compared with $F_0$ treatment. The GWP indicated significant variation between various cultivation and nitrogen application measures (Table 5). During the two-year study, $P_{ON75\%+DCD}$ produced considerable maximum-scale GWP production compared to all other processes. The average value of the data indicated that the output scale GWP of $P_{FN}$, $P_{ON}$, $P_{ON75\%+DCD}$, $P_{ON75\%+NC}$, $F_{FN}$, $F_{ON}$, $F_{ON75\%+DCD}$, and $F_{ON75\%+NC}$ treatments increased by 26.7%, 86.9%, 73.8%, 94.4%, 82.5%, 8.3%, 47.6%, 79.2%, and 74.9% when compared with $F_0$ treatment.

**Table 4.** Characteristics of seasonal greenhouse gas fluxes, GWP, and GHGI in maize cropping fields under different treatments [a] during 2019–2020 growing seasons.

| Treatments | GHG Flux (kg ha$^{-1}$) | | | GWP (kg CO$_2$-eq ha$^{-1}$) | | | | GHGI |
|---|---|---|---|---|---|---|---|---|
| | CH$_4$ | N$_2$O | NECB | CH$_4$ | N$_2$0 | NECB | Net | (kg CO$_2$-eq kg$^{-1}$ Grain) |
| 2019 | | | | | | | | |
| P$_0$ | 1.0 d | 4.5 b | −2482 i | 21.3 f | 1334 e | −8998 f | 10,591 g | 0.9 d |
| P$_{FN}$ | 1.8 c | 5.2 a | −3833 e | 37.3 e | 1471 d | −13,955 d | 15,604 e | 1.9 c |
| P$_{ON}$ | 1.2 c | 4.7 b | −3605 f | 26.3 f | 1431 d | −12,915 e | 15,057 e | 1.6 c |
| P$_{ON75\%+DCD}$ | 2.7 a | 5.7 a | −4743 a | 63.3 c | 1707 a | −17,290 a | 19,165 a | 2.0 b |
| P$_{ON75\%+NC}$ | 2.2 b | 5.5 a | −4665 b | 48.3 d | 1633 b | −17,003 a | 18,750 b | 1.8 |
| F$_0$ | 1.6 c | 4.3 b | −1129 j | 42.3 d | 1289 f | −4040 g | 5944 h | 1.2 c |
| F$_{FN}$ | 2.4 b | 4.8 b | −3550 g | 61.6 c | 1409 d | −13,117 d | 14,673 f | 2.b |
| F$_{ON}$ | 2.1 b | 4.5 b | −3324 h | 45.9 d | 1349 e | −12,086 e | 13,787 g | 1.6 c |
| F$_{ON75\%+DCD}$ | 3.3 a | 5.1 a | −4135 c | 84.1 a | 1558 c | −15,060 b | 17,054 c | 3.3 a |
| F$_{ON75\%+NC}$ | 2.9 a | 4.9 b | −4034 d | 72.9 b | 1528 c | −14,688 c | 16,476 d | 2.4 b |
| 2020 | | | | | | | | |
| P$_0$ | 1.1 c | 3.8 c | −2431 i | 21.9 | 1158 e | −8821 h | 10,270 f | 1.0 e |
| P$_{FN}$ | 1.5 c | 4.5 b | −3700 e | 32.0 | 1349 c | −13,477 e | 15,036 c | 1.9 d |
| P$_{ON}$ | 1.3 c | 4.1 b | −3497 g | 31.2 | 1230 d | −12,728 f | 14,993 d | 1.5 d |
| P$_{ON75\%+DCD}$ | 2.4 b | 5.8 a | −4822 a | 64.5 | 1695 a | −17,578 a | 19,279 a | 2.4 c |
| P$_{ON75\%+NC}$ | 2.1 b | 5.4 a | −4658 b | 51.3 | 1621 a | −16,988 b | 19,029 a | 2.0 c |
| F$_0$ | 1.4 c | 3.7 c | −1161 j | 33.6 | 1111 e | −4165 i | 5913 g | 1.7 d |
| F$_{FN}$ | 3.3 a | 4.2 b | −3598 f | 83.0 | 1278 d | −13,102 e | 14,400 d | 3.0 b |
| F$_{ON}$ | 1.7 c | 4.0 b | −3043 h | 43.4 | 1218 d | −11,056 g | 12,517 e | 2.4 c |
| F$_{ON75\%+DCD}$ | 3.9 a | 4.9 b | −4128 c | 98.0 | 1546 b | −15,045 c | 17,033 b | 4.5 a |
| F$_{ON75\%+NC}$ | 3.6 a | 4.5 b | −3933 d | 90.5 | 1356 c | −14,317 d | 15,898 c | 3.1 b |

**Table 5.** Effects of different treatments [a] on area and yield-scaled GWP, biomass yield, grain yield, evapotranspiration (ET), water use efficiency (WUE), and nitrogen use efficiency (NUE) of maize during 2019–2020 growing seasons.

| Treatments | Area-Sealed GWP (kg CO$_2$-eq ha$^{-1}$) | Yield-Sealed GWP (kg CO$_2$-eq kg$^{-1}$) | Biomass Yield (t ha$^{-1}$) | Grain Yield (t ha$^{-1}$) | ET (mm) | WUE (kg ha$^{-1}$ mm$^{-1}$) | NUE (kg kg$^{-1}$) |
|---|---|---|---|---|---|---|---|
| 2019 | | | | | | | |
| P$_0$ | 170.9 f | 0.05 e | 14.5 e | 6.8 e | 274.5 e | 24.8 b | -- |
| P$_{FN}$ | 242.7 d | 0.13 d | 16.7 c | 9.2 b | 409.0 a | 22.5 c | 8.28 e |
| P$_{ON}$ | 221.4 e | 0.17 d | 16.0 c | 7.8 d | 339.6 d | 23.0 c | 4.35 f |
| P$_{ON75\%+DCD}$ | 308.9 a | 0.76 a | 19.9 a | 11.1 a | 382.3 b | 29.0 a | 24.93 a |
| P$_{ON75\%+NC}$ | 294.4 b | 0.28 c | 18.3 b | 9.9 b | 373.7 b | 26.5 b | 17.97 c |
| F$_0$ | 129.4 h | 0.03 e | 12.7 g | 6.1 f | 328.4 d | 18.6 d | -- |
| F$_{FN}$ | 150.2 g | 0.04 e | 14.4 e | 7.3 d | 436.9 a | 16.7 e | 4.14 f |
| F$_{ON}$ | 148.4 g | 0.06 e | 13.5 f | 6.9 e | 361.3 c | 19.1 d | 3.48 f |
| F$_{ON75\%+DCD}$ | 275.8 c | 0.44 b | 16.7 c | 9.8 b | 378.6 b | 25.9 b | 21.45 b |
| F$_{ON75\%+NC}$ | 231.4 d | 0.21 c | 15.3 d | 8.3 c | 364.9 c | 22.7 c | 12.75 d |
| 2020 | | | | | | | |
| P$_0$ | 160.3 f | 0.10 d | 15.1 f | 6.4 f | 251.5 e | 25.4 b | -- |
| P$_{FN}$ | 262.4 c | 0.71 b | 17.6 c | 9.3 c | 452.6 a | 20.5 c | 10.00 e |
| P$_{ON}$ | 230.3 d | 0.25 c | 15.8 e | 7.7 e | 342.2 c | 22.5 c | 5.65 g |
| P$_{ON75\%+DCD}$ | 339.8 a | 1.19 a | 20.7 a | 11.3 a | 405.7 b | 27.9 a | 28.41 a |
| P$_{ON75\%+NC}$ | 303.8 b | 0.35 c | 19.0 b | 10.7 b | 386.0 b | 27.7 a | 24.93 b |
| F$_0$ | 94.3 h | 0.08 d | 14.4 g | 6.0 g | 292.2 d | 20.5 c | -- |
| F$_{FN}$ | 156.8 f | 0.15 d | 17.0 c | 7.9 e | 495.9 a | 15.9 e | 6.55 g |
| F$_{ON}$ | 127.3 g | 0.09 d | 16.3 d | 6.9 f | 388.1 b | 17.8 d | 3.91 h |
| F$_{ON75\%+DCD}$ | 211.0 e | 0.23 c | 17.4 c | 9.8 c | 464.4 a | 21.1 c | 22.03 c |
| F$_{ON75\%+NC}$ | 209.6 e | 0.35 c | 16.5 d | 8.4 d | 448.6 a | 18.8 d | 14.17 d |

*3.6. Resources Use Efficiencies, Carbon Efficiency Ratio (CER), and Maize Production*

During 2019–2020, different cultivation and nitrogen management practices have considerably enhanced biomass and grain yield, as well as CER and resource use efficiencies (Table 5). Compared with the $F_0$ treatment, the $P_{ON75\%+DCD}$ treatment significantly enhanced (41.0%) the biomass yield. The mean biomass yield was significantly enhanced in the $P_0$, $P_{FN}$, $P_{ON}$, $P_{ON75\%+DCD}$, $P_{ON75\%+NC}$, $F_{FN}$, $F_{ON}$, $F_{ON75\%+DCD}$, and $F_{ON75\%+NC}$ treatments by 2.8%, 19.1%, 10.4%, 41.0%, 29.5%, 5.9%, 14.7%, 13.5%, and 14.6% compared to that of the $F_0$ treatment. The CER was the maximum (0.99) in the $P_{ON75\%+DCD}$ treatment, followed by the $P_{ON75\%+NC}$ treatment (0.94), and then under the $P_{FN}$ treatment (0.93). The lowest (0.71) CER was recorded in the $F_0$ treatment. Compared with the $F_0$, the mean grain yield with $P_0$, $P_{FN}$, $P_{ON}$, $P_{ON75\%+DCD}$, $P_{ON75\%+NC}$, $F_{FN}$, $F_{ON}$, $F_{ON75\%+DCD}$, and $F_{ON75\%+NC}$ treatments was significantly increased by 0.8%, 39.1%, 16.5%, 68.4%, 54.9%, 11.3%, 25.6%, 36.1%, and 27.0%, respectively (Table 5). The data showed that WUE with $P_0$, $P_{FN}$, $P_{ON}$, $P_{ON75\%+DCD}$, $P_{ON75\%+NC}$, $F_{FN}$, $F_{ON}$, $F_{ON75\%+DCD}$, and $F_{ON75\%+NC}$ treatments were considerably improved by 34.8%, 15.6%, 22.1%, 52.7%, 45.6%, 5.9%, 17.2%, 17.7%, and 1.1%, compared with $F_0$ treatment. While the NUE with $P_{FN}$, $P_{ON}$, $P_{ON75\%+DCD}$, and $P_{ON75\%+NC}$ treatments were significantly enhanced by 41.5%, 26.1%, 18.5%, and 37.2%, respectively, compared with $F_{FN}$, $F_{ON}$, $F_{ON75\%+DCD}$, and $F_{ON75\%+NC}$.

## 4. Discussion

*4.1. Effects of N management Practices on ET and SWS*

Mulching with different nitrogen management practices is usually used as a useful cultivation technique to enhance rain-fed maize yields by increasing soil moisture conditions [10]. In contrast, mulching with different nitrogen management measures significantly increased greenhouse gas emissions [24] and consumed soil carbon pools [39,40]. The use of plastic mulching and different nitrogen management measures to enhance crop production is still under debate. $P_{ON75\%+DCD}$ treatment can decrease drought. In the $P_{ON75\%+DCD}$ treatment, the SWS of maize was considerably higher than in the $F_{ON75\%+DCD}$ treatment. A number of studies have shown that nitrogen application can increase soil absorption of water and nitrogen content [41]. Unnecessary fertilizer use may lead to high water efficiency [11]. There is a positive correlation between crop yield and field evapotranspiration [5]. Ma et al. [42] revealed a considerable improvement in the ET due to low N supply and high soil water availability. In our research, we found that compared with $F_{FN}$ and $P_{FN}$ treatments, $P_{ON75\%+DCD}$ and $P_{ON75\%+NC}$ treatments with different nitrogen management measures resulted in lower total ET due to maximum soil evaporation. Oenema et al. [43] reported that, compared to the control plot, the plastic film with a low N level maintained maximum water conditions with a low total ET.

*4.2. Effects of N management Practices on Greenhouse Gas Emissions*

Changes in soil water storage and humidity conditions caused by mulching affected soil microbial populations and activities [44], the mineralization process [27], and soil absorption of $CH_4$ [45]. Regarding the $CH_4$ emission under the cover of plastic film, Tan et al. [46] all believe that plastic film covering reduces $CH_4$ absorption or increases $CH_4$ emissions. However, in our research, corn fields are used as sinks for $CH_4$ emissions. Soil carbon has a greater role in regulating the $CO_2$ flux from the soil and other climatic factors that favor microbial processes [47]. The increase in temperature under the film cover can stimulate microbial activity, thereby accelerating organic matter decomposition [48,49], which explains the increase in $CO_2$ flux. Compared with the $F_{FN}$ treatment, the $CO_2$ emissions of the $F_{ON75\%+DCD}$ and $F_{ON75\%+NC}$ treatments were significantly greater. The $N_2O$ emission is significantly lower under the $F_0$ treatment. Under the $P_{ON75\%+DCD}$ and $P_{ON75\%+NC}$ treatments, the $N_2O$ emissions are significantly increased compared to the $P_{FN}$ and $F_{FN}$ treatments, which is consistent with the findings of Ma et al. [18]. Li et al. [50] also pointed out that DCD is more effective in suppressing early $N_2O$ emissions from paddy

fields. Other studies report that adding DCD to nitrogen fertilizer cannot only reduce soil $N_2O$ emissions by 39% [51] but also significantly reduce $N_2O$ emissions from rice fields [50].

### 4.3. Effects of N management Practices on GWP, GHGI, and CMI

The MBC ranges from 113.7 to 414.6 mg $kg^{-1}$. Under the $P_{ON75\%+DCD}$ treatment, the accumulation of MBC was significantly higher (400.3 mg $kg^{-1}$) compared to other treatments. Compared with $F_0$ treatment, the application of $P_{ON75\%+DCD}$ treatment showed a significant increase of 67% in MBC, respectively. Compared to other treatments, the $P_{ON75\%+DCD}$ treatment considerably improved the TC content (5.08 g $kg^{-1}$). The content of easily mineralizable carbon (RMC) was the highest in the plots treated with $P_{ON75\%+DCD}$ and $P_{ON75\%+NC}$ (177.7–137.9 mg $kg^{-1}$) and the lowest under the $F_0$ treatment (25.5 mg $kg^{-1}$). The microbial biomass in the soil is often dynamic when nutrient utilization is limited [52]. In this case, with the enhancement of SOC mineralization, the net soil carbon loss increases. Due to the limited availability of nitrogen and net immobilization, straw with a high C:N ratio tends to slowly decompose [53]. It was found that the total N under the $P_{ON75\%+DCD}$ and $P_{ON75\%+NC}$ treatments was significantly higher (0.61–0.57 g $kg^{-1}$) than that of all other treatments. Compared with the $F_{ON75\%+DCD}$ and $F_{ON75\%+NC}$ treatments, the C:N ratio was significantly higher (8.55–8.37) under the $P_{ON75\%+DCD}$ and $P_{ON75\%+NC}$ treatments. A single or combined application of inorganic fertilizers may produce more unstable carbon, which can be used as a source of nutrients [29]. The CMI was considerably improved by 31.2%, 10.2%, 11.4%, 10.8%, 10.8%, and 13.4% under the treatments of $P_0$, $P_{FN}$, $P_{ON}$, $P_{ON75\%+DCD}$, and $P_{ON75\%+NC}$, which was higher than that of $F_0$, $F_{FN}$, $F_{ON}$, $F_{ON75\%+DCD}$, and $F_{ON75\%+NC}$ treatments. These planting method data are similar to those reported by Whitbread et al. [54].

### 4.4. Effects of N management Practices on Resource Use Efficiency and Maize Production

The $(NO_3^--N)$ under $F_0$ treatment was significantly lower than all other treatments. Under the treatments of $P_{ON75\%+DCD}$ and $P_{ON75\%+NC}$, $(NO_3^--N)$ significantly increased compared with PFN and FFN treatments. Compared with F0, the $NH_4^+-N$ of all other treatments was significantly increased, while the $NH_4^+-N$ of the $P_{ON75\%+DCD}$ and $P_{ON75\%+NC}$ treatments had no significant changes under the two cultivation methods. This may be due to the inhibitory effect of DCD on ammonia-oxidizing bacteria and related enzymes, effectively delaying the oxidation process of $NH_4^+-N$ to $NO_3^--N$ [37,55]. By adjusting the rapid conversion of soil nitrogen and maintaining a high soil $NH_4^+-N$, the accumulation and leaching loss of $NO_3^--N$ can be effectively reduced, and $N_2O$ emissions can be reduced [56]. Considering that nitrogen management practices and mulching films are widely used in arid regions [57]. The GWP data on the agricultural system can provide information on the impact of agricultural practices on climate change [3]. In our research, the treatments of $P_{ON75\%+DCD}$ and $P_{ON75\%+NC}$ with mulching film significantly reduced GHGI compared to traditional flat-land cultivation because the yield of corn was greatly increased. GHGI is a potential barometer to compare the impact of global warming on agricultural management and crop yields [48]. This increase in GHGI is mainly due to the massive consumption of soil carbon storage rather than an improvement in GHGI. Current research has indicated that improving crop yields can effectively reduce GHGI [58]. Compared with the $P_{FN}$ and $F_{FN}$ treatments, the $P_{ON75\%+DCD}$ treatment improved the net GWP, which was mainly due to the significant increase in soil carbon pool consumption. Current research has shown that increasing corn yield can decrease GHGI [49].

Compared with the $F_0$ treatment, the $P_{ON75\%+DCD}$ treatment significantly increased (41.0%) biomass yield. Plastic mulching with reasonable nitrogen application effectively utilizes rainfall; therefore, compared with flat planting, it increases grain yield with a higher WUE [59]. Compared with $F_0$ treatment, the average grain yield of $P_0$, $P_{FN}$, $P_{ON}$, $P_{ON75\%+DCD}$, $P_{ON75\%+NC}$, $F_{FN}$, $F_{ON}$, $F_{ON75\%+DCD}$, and $F_{ON75\%+NC}$ treatments was significantly increased by 0.8%, 39.1%, 16.5%, 68.4%, 54.9%, 11.3%, 25.6%, 36.1%, and 27.0%. WUE shows the link between water use and crop productivity. Liu et al. [37] also investigated

that the optimum fertilizers under plastic mulching increased grain yield and reduced the ET; therefore, the rainwater with high WUE and NUE was effectively used. Soil fertility status considerably affects resource utilization efficiencies. Low N levels result in higher NUE, while high N levels result in lower NUE [60]. Taking into account the impact on greenhouse gas emission reduction, corn yield response, and greenhouse gas emission factors, $P_{ON75\%+DCD}$ treatment can be suggested as the preferred cultivation and nitrogen management practice for increasing yield and coping with climate change.

## 5. Conclusions

In a semi-arid agricultural ecosystem, the application of plastic mulch under the ridge cropping system reduces the optimal N + dicyandiamide by 25%, resulting in soil carbon buildup and increased corn yield. The results showed that compared to other treatments, $P_{ON75\%+DCD}$ significantly increased SWS, WUE, and NUE because the total ET and GHG emissions were reduced. Under $P_{ON75\%+DCD}$ or $P_{ON75\%+NC}$, the soil carbon storage significantly increased. The $P_{ON75\%+DCD}$ treatment is more effective in improving MBC, CMI, and WSC, although it increases gaseous carbon emissions more than all other treatments. Compared with FFN, under the $P_{ON75\%+DCD}$ treatment, the overall $CH_4$, $N_2O$, and $CO_2$ emissions are all reduced. Under the $P_{ON75\%+DCD}$ treatment, the area scale GWP (52.7%), yield scale GWP (90.3%), biomass yield (22.7%), WUE (42.6%), NUE (80.0%), and grain yield (32.1%) are improved instead of FFN, which might offset the negative ecological impacts. The $P_{ON75\%+DCD}$ treatment can bring obvious benefits in terms of increasing yield, reducing global warming, and maintaining soil health.

**Author Contributions:** H.R. conceived and designed the experiments. H.R., S.X., F.Z., M.S. and R.Z. performed the experiments. H.R. and S.X. analyzed the data and wrote the manuscript. All authors have read and agreed to the published version of the manuscript.

**Funding:** Scientific Research Business Expenses of Heilongjiang Scientific Research Institutes (Grant No. CZKYF2022-1-B024; Grant No. CZKYF2022-1-C008).

**Data Availability Statement:** Data will be available upon personal request.

**Conflicts of Interest:** The authors declare no conflict of interest.

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
