# Peer review of "Cultivation and Nitrogen Management Practices Effect on Soil Carbon Fractions, Greenhouse Gas Emissions, and Maize Production under Dry-Land Farming System"

_land, doi:10.3390/land12071306_

Round 1

Reviewer 1 Report

Dear Authors

The study is quite interesting and useful data have been generated. However, a complete reformatting of the paper is required as many sentences are not clear and not giving appropriate meaning. The authors can take average of both year data or must have to mention the year while presenting the observation which is lacking. In some cases, the data presented in the table and in text is not matching which needs to be presented carefully.

Line 14: delete ‘was’

Line 16-17: You can write two planting methods during 2019-20: P: plastic mulching ridges; F: traditional flat planting with five nitrogen management practices i.e.0: no nitrogen fertilizer; FN: a common nitrogen fertilizer rate for farmers 290 17 kg ha-1………

Line 38-39: From 2013 to 2019, the plastic film mulching global demand is expected to increase by 7.6%. Expected to increase or increased?

Line 54: How the excessive use of nitrogen and irrigation water cause reduction in rain water?

Amount of phosphorus and potassium applied to be mentioned. Management practices such as tillage, intercultural operation, irrigation interval etc. to be mentioned.

Line 122-124: The sentences may be changed. Work done should be mentioned rather than the method described in textbook or manual. Sentences may be in past participle form

Which design has been followed for the field experiment?

Line 186-187: Sentence is not clear

Line 210: Check the value of MBC

Line 211-212: respective to?

Moderate English editing is required. Many sentences are not clear.

Author Response

Thank you to review our manuscript! You are kind and responsible reviewers, and the suggestions you have given are all valuable and very helpful for revising and improving our paper. We are very grateful for that. We have studied your comments carefully and have made corrections; Many thanks to the Editor and the Reviewers for your time and thoughtful comments, many of which have been incorporated into the revised manuscript.

Below are our detailed responses (in BOLD type) to Editor and Reviewer’s comments, (the page and line numbers refer to our revised manuscript):

Dear Authors

The study is quite interesting and useful data have been generated. However, a complete reformatting of the paper is required as many sentences are not clear and not giving appropriate meaning. The authors can take average of both year data or must have to mention the year while presenting the observation which is lacking. In some cases, the data presented in the table and in text is not matching which needs to be presented carefully.

Response: respected reviewer, thank you very much for your encouragement and excellent suggestions. Sir, I extremely work hard to improve my paper with your suggestions, sir I also cross-checked my data in the table and text and removes an error, and presents it carefully, please see in new revised manuscript.

Line 14: delete ‘was’

Response: We are sorry about that. According to your suggestion, I delete the word “was”; please see in the new revised manuscript.

Line 16-17: You can write two planting methods during 2019-20: P: plastic mulching ridges; F: traditional flat planting with five nitrogen management practices i.e.0: no nitrogen fertilizer; FN: a common nitrogen fertilizer rate for farmers 290 17 kg ha-1………

Response: We are sorry about that. I revise the above sentence according to your suggestion; please see in the new revised manuscript.

Line 38-39: From 2013 to 2019, the plastic film mulching global demand is expected to increase by 7.6%. Expected to increase or increased?

Response: respected reviewer, thank you very much for your excellent suggestion; sir I correct the expected to increased, please see the new revised manuscript.

Line 54: How the excessive use of nitrogen and irrigation water causes reduction in rain water?

Response: We are sorry about that. Sir, it’s a typing mistake the excessive use of nitrogen and irrigation water causes a reduction in crop production, not a rainwater, please see in the new revised manuscript.

Amount of phosphorus and potassium applied to be mentioned. Management practices such as tillage, irrigation interval etc. to be mentioned.

Response: We are sorry about that. According to your above suggestion, I mentioned the recommended doses of P and K at 90 and 60 kg ha-1 apply one day before sowing. During both growing seasons, irrigation was not supplied, while conventional tillage practice was used for soil flow, and manually weeds were controlled; please see in the new revised manuscript.

Line 122-124: The sentences may be changed. Work done should be mentioned rather than the method described in textbook or manual. Sentences may be in past participle form

Response: respected reviewer, thank you very much for your excellent suggestion; I correct the above sentence according to your suggestion, and remove the grammar mistakes, please see in the new revised manuscript.

Which design has been followed for the field experiment?

Response: respected reviewer, thank you very much for your excellent suggestion; I follow the randomized completely block design were used having three replications, please see in the new revised manuscript.

Line 186-187: Sentence is not clear

Response: respected reviewer, thank you very much for your excellent suggestion; I correct the above sentence according to your suggestion, and remove the grammar mistakes, please see in the new revised manuscript.

Line 210: Check the value of MBC

Response: respected reviewer, thank you very much for your excellent suggestion; sir I check the value of MBC and found it correct, the MBC value is 400.3, sir I get this value by taking an average of both year of study (386.1+ 414.6 / 2) = (400.3), also sir in my article most of the value is the average of the two-year study, sir please see in the new revised manuscript.

Line 211-212: respective to?

Response: We are sorry about that. Sir, I deleted the respective to according to your suggestion, please see in the new revised manuscript.

Comments on the Quality of English Language

Moderate English editing is required. Many sentences are not clear.

 Response: Thank you, following your suggestion, I corrected all grammar mistakes according to your above suggestion, sir we have also sent our manuscript to a professional, native English-speaking Scientific Editor to improve the language and specifically to remove grammar mistakes. Sir certificate for the concerned company is also attached to my paper.

Reviewer 2 Report

The manuscript falls within the scope of the LAND journal and presents findings from an experiment comparing two planting methods and five N management techniques. The manuscript is well-written with few errors in English and grammar. Please consider the following detailed comments for improvement:

  1. Replace the word "These" with a specific description of the planting methods.
  2. Expand the abbreviations at their first mention for better clarity.
  3. Clarify whether it is the ET rate or total ET being referred to.
  4. Provide a quantitative figure to indicate the extent of carbon storage increase.
  5. Expand the acronym "PFM" at its first mention.
  6. Specify the global demand being referred to.
  7. Clarify the meaning of "CPI" in the context.
  8. The figures and tables present the dataset adequately.
  9. Spell out the symbols used in all equations.
  10. Provide a detailed description of how the flux was measured in the methodology section.
  11. Define the meaning of "R" and "D" in the manuscript.
  12. Write the equations in complete form, including the units of measurement. The same applies to NUE.
  13. Indicate the depth of soil water storage referred to in Figure 2.
  14. Include the equation for yield-scaled GWP in the methodology section.
  15. Clarify the meanings of the acronyms used in equations 1 and 2.
  16. Confirm whether "GHGI" stands for Greenhouse Gas Intensity and provide clarification.
  17. Organize the discussion section into subsections that align with the study's aims or results, using appropriate subtitles. Please consider this study. https://doi.org/10.1111/ejss.13124

With moderate changes, the article can be suitable and acceptable for publication. Please consider addressing these comments to enhance the manuscript.

 Minor editing of English language required

Author Response

Thank you to review our manuscript! You are kind and responsible reviewers, and the suggestions you have given are all valuable and very helpful for revising and improving our paper. We are very grateful for that. We have studied your comments carefully and have made corrections; Many thanks to the Editor and the Reviewers for your time and thoughtful comments, many of which have been incorporated into the revised manuscript.

Below are our detailed responses (in BOLD type) to Editor and Reviewer’s comments, (the page and line numbers refer to our revised manuscript):

The manuscript falls within the scope of the LAND journal and presents findings from an experiment comparing two planting methods and five N management techniques. The manuscript is well-written with few errors in English and grammar. Please consider the following detailed comments for improvement:

 Response: respected reviewer, thank you very much for your encouragement and excellent suggestions. Thank you for your advice, sir I extremely work hard to improve my paper with your suggestions, please see in new revise the manuscript.

I corrected all grammar mistakes according to your above suggestion, sir we have also sent our manuscript to a professional, native English-speaking Scientific Editor to improve the language and specifically to remove grammar mistakes. Sir certificate for the concerned company is also attached to my paper.

  1. Replace the word "These" with a specific description of the planting methods.

Response: We are sorry about that. I replace the word “These” according to your above suggestion; please see in the new revised manuscript.

  1. Expand the abbreviations at their first mention for better clarity.

Response: We are sorry about that. According to your suggestion, I expand all the abbreviations at first time mentioned, please see in the new revised manuscript.

  1. Clarify whether it is the ET rate or total ET being referred to.

Response: Thank you very much for the suggestion, sir it's total ET, sir please see in the whole article its total ET.

  1. Provide a quantitative figure to indicate the extent of carbon storage increase.

Response: We are sorry about that. According to your above suggestion, I add a quantitative figure to indicate the extent of carbon storage increase in Table 2; please see in the new revised manuscript.

  1. Expand the acronym "CMI" at its first mention.

Response: We are sorry about that. CMI stands for carbon management index.

  1. Specify the global demand being referred to.

Response: We are sorry about that. According to your above suggestion, I add the Specify the global demand being referred to in the introduction section; please see in the new revised manuscript.

  1. Clarify the meaning of "CPI" in the context.

Response: We are sorry about that. CPI stands for carbon pool index.

  1. The figures and tables present the dataset adequately.

Response: respected reviewer, thank you very much for your encouragement and excellent suggestions.

  1. Spell out the symbols used in all equations.

Response: We are sorry about that. According to your suggestion, I spell out the symbols of all equations, please see in the new revised manuscript.

  1. Provide a detailed description of how the flux was measured in the methodology section.

Response: We are sorry about that. According to your suggestion, I rewrite the above methodology in full detail and clearly explain each and every point, please see in the new revised manuscript.

  1. Define the meaning of "R" and "D" in the manuscript.

Response: Thank you very much for this suggestion, where R is the daily gas emission rates, D is the number of days between the ith sampling interval.

  1. Write the equations in complete form, including the units of measurement. The same applies to NUE.

Response: We are sorry about that. According to your suggestion, I rewrite the above equation in a complete manner with full mention units just like NUE please see in the new revised manuscript.

  1. Indicate the depth of soil water storage referred to in Figure 2.

Response: Thank you very much for this suggestion. The soil water storage is measured up to 0-120 cm depth please see in figure 2.

  1. Include the equation for yield-scaled GWP in the methodology section.

Response: We are sorry about that. According to your suggestion, I mention the yield scaled GWP equation in the methodology section; please see in the new revised manuscript.

  1. Clarify the meanings of the acronyms used in equations 1 and 2.

Response: We are sorry about that. According to your suggestion, I spell out the symbols of equations 1-2, please see in the new revised manuscript.

  1. Confirm whether "GHGI" stands for Greenhouse Gas Intensity.

Response: Thank you very much for this suggestion, yes sir it is Greenhouse gas intensity (GHGI).

  1. Organize the discussion section into subsections that align with the study's aims or results, using appropriate subtitles. Please consider this study. https://doi.org/10.1111/ejss.13124

Response: respected reviewer, thank you very much for your encouragement, excellent suggestions, and comments. I divide the discussion section with subtitles, also I get great help from your above mention article.  Sir please sees in new revise article.

With moderate changes, the article can be suitable and acceptable for publication. Please consider addressing these comments to enhance the manuscript.

Response: respected reviewer, thank you very much for your encouragement, excellent suggestions and comments.

Comments on the Quality of English Language

 Minor editing of English language required

 Response: Thank you, following your suggestion, I corrected all grammar mistakes according to your above suggestion, sir we have also sent our manuscript to a professional, native English-speaking Scientific Editor to improve the language and specifically to remove grammar mistakes. Sir certificate for the concerned company is also attached to my paper.

Reviewer 3 Report

The experiment design was not clearly mentioned in this study

English Language standard is statisfactory.

Author Response

Thank you to review our manuscript! You are kind and responsible reviewers, and the suggestions you have given are all valuable and very helpful for revising and improving our paper. We are very grateful to that. We have studied your comments carefully and have made correction; Many thanks to the Editor and the Reviewers for your time and thoughtful comments, many of which have been incorporated into the revised manuscript.

Below are our detailed responses (in BOLD type) to Editor and Reviewer’s comments, (the page and line numbers refer to our revised manuscript):

The experiment design was not clearly mentioned in this study

Response: We are sorry about that. According to your above suggestion, I add the experiment design in detail, A randomized completely block design was used having three replications; please see the newly revised manuscript.

English Language standard is satisfactory.

 Response: Thank you, following your suggestion, I corrected all grammar mistakes according to your above suggestion, sir we have also sent our manuscript to a professional, native English-speaking Scientific Editor to improve the language and specifically to remove grammar mistakes. Sir certificate for the concerned company is also attached to my paper.

Round 2

Reviewer 3 Report

There is mistake of formula of ET in line no. 120-121. Either correct it or remove it.

Satisfactory

Author Response

Thank you to review our manuscript! You are kind and responsible reviewers, and the suggestions you have given are all valuable and very helpful for revising and improving our paper. We are very grateful to that. We have studied your comments carefully and have made correction; Many thanks to the Editor and the Reviewers for your time and thoughtful comments, many of which have been incorporated into the revised manuscript.

Below are our detailed responses (in BOLD type) to Editor and Reviewer’s comments, (the page and line numbers refer to our revised manuscript):

Comments and Suggestions for Authors

There is mistake of formula of ET in line no. 120-121. Either correct it or remove it.

Response: We are sorry about that. According to your above suggestion, I remove the ET formula; please see in the new revised manuscript.
